# Evaluation of molecular characterization and phylogeny for quantification of *Acanthamoeba* and *Naegleria fowleri* in various water sources, Turkey

**Mehmet Aykur**[1,2]*, **Hande Dagci**[2]

1 Department of Parasitology, Faculty of Medicine, Tokat Gaziosmanpasa University Tokat, Tokat, Turkey,
2 Department of Parasitology, Faculty of Medicine, Ege University, Bornova, İzmir, Turkey

* mehmetaykur@gmail.com

**Data Availability Statement:** All relevant data are within the paper and its Supporting Information files.

## Abstract

Free-living amoeba (FLA) is widely distributed in the natural environment. Since these amoebae are widely found in various waters, they pose an important public health problem. The aim of this study was to detect the presence of *Acanthamoeba*, *B. mandrillaris*, and *N. fowleri* in various water resources by qPCR in Izmir, Turkey. A total of (n = 27) 18.24% *Acanthamoeba* and (n = 4) 2.7% *N. fowleri* positives were detected in six different water sources using qPCR with ITS regions (ITS1) specific primers. The resulting concentrations varied in various water samples for *Acanthamoeba* in the range of $3.2 \times 10^5$-$1.4 \times 10^2$ plasmid copies/l and for *N. fowleri* in the range of $8 \times 10^3$-$11 \times 10^2$ plasmid copies/l. The highest concentration of *Acanthamoeba* and *N. fowleri* was found in seawater and damp samples respectively. All 27 *Acanthamoeba* isolates were identified in genotype level based on the 18S rRNA gene as T4 (51.85%), T5 (22.22%), T2 (14.81%) and T15 (11.11%). The four positive *N. fowleri* isolate was confirmed by sequencing the ITS1, ITS2 and 5.8S rRNA regions using specific primers. Four *N. fowleri* isolates were genotyped (three isolate as type 2 and one isolate as type 5) and detected for the first time from water sources in Turkey. *Acanthamoeba* and *N. fowleri* genotypes found in many natural environments are straightly related to human populations to have pathogenic potentials that may pose a risk to human health. Public health professionals should raise awareness on this issue, and public awareness education should be provided by the assistance of civil authorities. To the best of our knowledge, this is the first study on the quantitative detection and distribution of *Acanthamoeba* and *N. fowleri* genotypes in various water sources in Turkey.

## Introduction

Free-living amoeba (FLA) are unicellular protozoa that commonly find in soil and water throughout the world. Free-living amoeba could be found in tap water, well water, seawater, streams, river, swimming pools, dams, lakes, and air-conditioning systems [1]. Among the

**Funding:** The research was supported by a grant from the Budget of the Academic Staff Training Program by The Council of Higher Education and the Scientific Research Projects Branch Directorate of Ege University, Turkey (Project No: 18-TIP-025). The funders had no role in study design, data collection and analysis, decision to publish, or preparation of the manuscript.

**Competing interests:** The authors have declared that no competing interests exist.

numerous FLA species present in nature, the most common species are *Acanthamoeba*, *Balamuthia mandrillaris (B. mandrillaris)*, and *Naegleria* species, which play a role in human and animal infections [2]. *Acanthamoeba* spp. and *B. mandrillaris* may cause granulomatous amoebic encephalitis (GAE), cutaneous lesions, lung infections, and also *Acanthamoeba* keratitis (AK) in immunocompetent persons. *Acanthamoeba* genus is divided into 22 different genotypes based on the 18S rRNA gene, and genotype T4 is one of the most common in the environment and the most common genotype causing human infection [3, 4]. *N. fowleri* causes primary amoebic meningoencephalitis (PAM) in immunocompetent children and young adults [1, 5].

This pathogenic FLA enters the body via nasal mucosa and/or the skin lesions and then disseminate along the olfactory neuroepithelial route or by following the hematogenous spread route they gain entry into the brain to occur infection [6]. Quantitative screening of these amoebae in various water sources is crucial since they pose risk to human health. To date, in vitro culture methods were used to quantitatively assessment of aquatic FLA, but there are some limitations such as time-consuming procedure, precision, and accuracy [7]. Quantitative real-time PCR (qPCR) assay is a method with high specificity and sensitivity useful for detecting the presence of the amoebae in water resources [8, 9].

The aim of this study was to identify rapidly and accurately the presence of *Acanthamoeba* spp., *B. mandrillaris*, and *N. fowleri* in various water sources by qPCR assay. Moreover, the sensitivity, specificity, and efficiency of the qPCR were also evaluated. Finally, the water quality parameters were measured and *Acanthamoeba* culture-positive samples were subjected to osmo/thermo-tolerance test to measure their pathogenicity. In the light of the results obtained, identified isolates were evaluated as potential risks for humans.

## Material and methods

### The geographical location of the study area

The study is conducted in the province of Izmir, which is the third-largest city in western Turkey. Izmir is located between the northern latitudes 37° 45' and 39° 15' and 26° 15' and the east longitudes 28° 20' and have a surface area of 12.012 km$^2$. Izmir city is in Mediterranean climate zone and summers are also hot and dry and followed by mild and wet winters. According to National Meteorological Service average temperature in summertime was higher than 30°C and highest temperature might be higher than 40°C. While the approximate population for 2020 was 4.394.694, this number increases even more during the summer months, since the region is one of the important touristic areas in Turkey. There are many irrigation dams, lakes, and ponds in this region.

### Water sample collection and processing

A total of 148 water samples were collected from places within the boundaries of Izmir that could pose a risk to people and where human contact was high. Tap water (TW), pool water (PW), well water (WW), lake water (LW), dam water (DW), stream water (StW), seawater (SeW), and thermal spring water (TsW) were collected from various water sources and the geographic coordinates were shown (S1 Table). Water samples were collected in approximately one liter (lt) of the sterile glass bottles and stored at 4°C for subsequent analyses within 24 hr.

For culture and DNA isolation one liter of water sample was concentrated by filtration using a nitrocellulose membrane with a pore size of 0.22 μm (Sartorius Stedim Biotech, Göttingen, Germany). The filter membrane was divided into two equal parts with the help of a sterile scalpel and forceps, then half of the filter was transferred into the center of the NNA plate [10].

## Analysis of water quality parameters

Water quality parameters, including total dissolved solids content (TDS), electrical conductivity (EC), and temperature were measured in situ using the portable thermometer (TDS&EC meter hold). The temperature, TDS, and EC of the parameters indicate sensitivity in degrees between 0.1 and 80.0˚C (Celsius), 0 and 5000 ppm (parts per million), 0 and 9990 μs/cm, respectively. The chlorine level of the water samples was evaluated in situ using the chlorine test kit (Sutest Liquid Test Kit). The pH of the collected water samples was determined using a pH-meter (HI 2211–02, Hanna Instruments Inc., Woonsocket, MA, USA) in Ege University Parasitology Department Laboratory.

## Culture of free-living amoeba

After filtration of each water sample, the half of the cut filter was placed in the center of 2% non-nutrient agar (NNA) plates previously seeded with heat killed 100 μl *Escherichia coli* (ATCC 25922) bacterial suspension and the edges of the plates were sealed around with parafilm (Heathrow Scientific, Vernon Hills, IL, USA). The plates were incubated in the inverted position at 30–32˚C and exanimated daily with the inverted microscope for 10 days. Plates without proliferation were considered negative after a check of at least two weeks. FLA-positive plates were then sub-cultured by cutting off small pieces circled with a pen under the microscope and transferring them to new fresh NNA plates to purifying them from other organisms, especially fungi and yeasts. The grown of *Acanthamoeba* trophozoites and cysts were characterized from other free-living amoebas. Besides, the cyst shape has been easily identified by the double wall of the cyst and typical star shape [11, 12].

## Tolerance assays for *Acanthamoeba* positive samples

Pathogenicity tests were repeated three times for each positive sample. The pathogen strains of *Acanthamoeba castellanii* and *Acanthamoeba* spp. (EU266547 –T4) from Cumhuriyet and Dokuz Eylül Universities were used as reference strains.

**Osmo-tolerance assay.** To investigate the effect of osmolarity of each isolate on the trophozoites of *Acanthamoeba* (approximately $10^3$ trophozoites/plate), trophozoites were coated with mannitol-free *E. coli* and transferred to the center of NNA plates (as a control). Positive isolates (approximately $10^3$ trophozoites/plate) were transferred to the center of the plates by coating the NNA plate with *E. coli* suspension prepared at 0.5 M and 1 M mannitol concentration. The plates were then incubated at 30˚C for 10 days and the growth of amoebae at 24, 48, and 72 hours was evaluated. Trophozoites or cysts were counted at a microscopic magnification at x100 of five microscopic areas of approximately 20 mm from the center of each plate. The presence of proliferation was evaluated as (+) positive, and the absence of growth (-) as negative [13, 14].

**Thermo-tolerance assay.** For the thermo-tolerance assay, the trophozoites of *Acanthamoeba* spp. (approximately $10^3$ trophozoites/plate) were transferred to the center of *E. coli* coated NNA plates. These plates were incubated at 30˚C (as a control), 37˚C, and 42˚C for 10 days; It was evaluated after 24, 48, and 72 hours of the incubation. During this period, proliferation was evaluated under the microscope as mentioned in the osmo-tolerance test [13, 14].

## DNA extraction from culture and filter membranes

For the DNA isolation of the amoeba from the NNA plates, which were identified as *Acanthamoeba* spp. by microscopy, 2 ml of 1xPBS (Thermo Fisher Scientific, Phosphate-Buffered Saline (PBS), pH: 7.4) buffer solution was dropped onto the plate. The amoebas from agar

plates were collected into the tube using the sterile swab after waiting for approximately 5 minutes. The tubes were centrifuged at 2500 rpm for 10 min and washed with PBS buffer solution. *Acanthamoeba* genomic DNA was extracted with the QIAamp DNA mini kit (Qiagen GmbH, Germany) according to the manufacturer's recommendations.

Half of the 0.22 μm diameter membrane of the filtered water sample was cut into several pieces with sterile scissors and transferred to the bead tube. Total genomic DNA was extracted using the Norgen Biotek Water RNA/DNA Purification Kit (Water RNA / DNA Norgen Biotek Corp., Canada) following the manufacturer's protocol. Briefly, after placing the filter into the tube with beads, 500 μl Lysis buffer E was added. The tubes were vortexed for 30 sec using FastPrep®-24 instrument (MP Biomedical). The tubes were then incubated in the thermal block at 65°C for 10 min. and then centrifuged at 20 000 g for 1 min. After centrifugation, 600 μl ethanol was added to the mixture and transferred to filter tubes. The filter tubes were then washed twice with 400 μl of wash solution A and genomic DNA sample was obtained by adding 100 μl of elution buffer H. DNA concentration and purity were measured using the NanoDrop® 1000 spectrophotometer (NanoDrop Technologies, Wilmington, DE, USA). DNA samples were kept at -20°C until the PCR experiments.

## Positive control plasmid for PCR

The positive control of *Acanthamoeba* spp. was obtained from a reference strain, which was isolated from a human case with *Acanthamoeba* keratitis (GenBank No: EU266547 –T4). DNA samples of *N. fowleri* and *B. mandrillaris* strains were obtained from the Center for Disease Control and Prevention (CDC).

The 18S rRNA gene for *Acanthamoeba* spp., and *B. mandrillaris* and the 5.8S rRNA and ITS (ITS1 and ITS2) regions for *N. fowleri* were selected as targets to determine the presence of the plasmid copy quantification in water samples [15–17]. The primers and conditions were used for amplification are the same as those used for the LightCycler 480 PCR test described below. A plasmid containing the PCR amplified product was commercially synthesized by Letgen Biotechnology Laboratory (Letgen Biotechnology, İzmir, Turkey) using the pGEM-T Vector cloning kit (Promega Corporation, Madison, WI) following the manufacturer's instructions. The number of copies in the plasmid solution was calculated using a NanoDrop ND1000 spectrophotometer (NanoDrop Technologies, Wilmington, DE, USA). Serial dilutions (plasmid controls ranging from $1x10^9$ to $1x10^0$ copies plasmid/μl) on the order of 10-fold of 18S rRNA gene and ITS region fragment were used to generate the standard curve of concentrations expressed in log units (log10) versus the values obtained in amplification cycles. Quantification analysis for each plasmid control was performed with the Light Cycler 480 II® Thermal Cycler (Roche Diagnostic) on 96-well white LightCycler 480® multiwell plates (Roche Diagnostics Ltd, Switzerland).

## Quantitative real-time PCR (qPCR) assay

The qPCR was performed using DNA obtained from cultures and direct water filters. The quantification of *Acanthamoeba* spp., *B. mandrillaris*, and *N. fowleri* DNA was performed by using a LightCycler 480 II (Roche Diagnostics, Mannheim, Germany) Real-Time PCR Systems. The targets sequence of the 180-bp (*Acanthamoeba* spp), 171-bp (*B. mandrillaris*), and 123-bp (*N. fowleri*) fragments of the 18S rRNA and ITS gene regions were amplified. The primers and the respective probes used in this study were described in S2 Table [15, 17]. Reaction was performed in a final volume of a 20 μl in 96-well plates, containing 5 μl DNA template or controls, 5 x TaqMan Master Mix (Roche), 0.25 μM each of primer and 0.2 μM of each probe and then centrifuged at 1500 g for 2 min at 4°C. The PCR conditions were conducted

using the following calculated control protocol: 10 min pre-incubation step at 95˚C, followed by 45 cycles of 10 sec at 95˚C, 1 min at 60˚C and 1 sec at 72˚C and followed by a final cooling step at 40˚C for 30 sec.

All qPCR reactions were run using positive control (plasmid DNA), negative control DNA (using double distilled water) and were tested in triplicate in each reaction. The *Ct* value (cycle threshold) was defined as the number of cycles required for the fluorescence signal to cross the threshold. The *Ct* value is inversely correlated with the quantity of DNA. A standard curve was constructed using a series of dilutions with a known quantity of plasmid DNA standards. The slope (*S*) of the standard curve was adopted as an indication of the efficiency of the real-time PCR amplification. The efficiency (*E*) of the qPCR amplification was calculated according to the equation of $E = 10^{(-1/Slope)}-1$.

## Assessment of possible the qPCR inhibition in water samples

Duplicate PCR reactions were performed to eliminate possible PCR inhibition that might arise from water samples. One of the reactions contained only purified water sample DNA, while the other reaction contained ten plasmid copies with the spiked into the purified water DNA sample. To generate a cycle threshold value, three reactions containing ten plasmid control DNA copies in distilled water were performed. The *Ct* values observed for the spiked water sample with a positive control plasmid differed from the mean >40 Ct value were considered to be indicative of PCR inhibition. The inhibited samples were diluted 10-fold and PCR was analyzed again as described above.

## DNA sequencing and phylogenetic analysis

To genotype qPCR positive samples, conventional PCR was performed for *Acanthamoeba* spp. and *Naegleria* spp., with primers specific for 18S rRNA gene (JDP1-JDP2) and ITS region (FW2-RV2), respectively (S2 Table) [18, 19]. PCR was performed in a total volume of 50 μl including 25 μl of 2x PCR Master Mix, 10 pmol of each primer and 5 μl template DNA. reaction was carried out in a Techne TC-3000 Thermal Cycle (Techne, Staffordshire, UK) by following program: an initial denaturation at 95˚C for 2 min followed by 35 cycles 95˚C for 35 s, annealing step at 60˚C and 58˚C for 40 s for *Acanthamoeba* spp. and *Naegleria* spp. respectively and final extension 3 min at 72˚C. All PCR products were separated by electrophoresis on 2% agarose gel, stained with Safeview Classic (Applied Biological Materials Inc., Richmond, Canada) and were photographed using an Alpha Imager HP (Alpha Innotech), on a UV transilluminator. All the PCR products were purified with the QIAquick PCR purification kit (QIAGEN, Germany) according to the manufacturer's instructions. Sequencing was performed by a commercial company using an ABI automated sequencing system (Microsynth, Balgach, Switzerland). The resulting sequences were alimented using ClustalW based on sequence analysis of DF3 and ITS region as previously described [20, 21] by comparing to the available *Acanthamoeba* and *N. fowleri* DNA sequences in GenBank database (Department of Molecular Genetics, The Ohio State University, OH, USA). Phylogenetic relationship among the sequences was made using the neighbor-joining with molecular distances under the Kimura two-parameter distance model with the Molecular Evolutionary Genetics Analysis (MEGA X) software program [22]. The accuracy of the phylogenetic tree was assessed by 1000 bootstrap replicate data sets. The root of the tree was established using an outgroup (*Saccamoeba lacustris* GenBank No: JN112797.1). Sequence data obtained for *Acanthamoeba* and *N. fowleri* isolates were deposited in the GenBank database under the accession numbers MW689474–MW689500 and MW677627-MW677629, MW676178, respectively.

## Statistical analysis

The data obtained were analyzed using Mac OSX SPSS 25.0 version (SPSS Inc, Chicago, IL, USA) software. Numerical variables were summarized using the mean and standard error of the mean. Kolmogorov-Smirnov test was used to determine the importance of normality. The Mann-Whitney U test was used to compare the relationship between water quality parameters and the presence/absence of *Acanthamoeba* spp., and *N. fowleri* in environmental water samples. A value of $p \leq 0.05$ was considered statistically significant.

## Results

### Isolation of *Acanthamoeba* spp. in culture

A total of 148 water samples were collected from 19 different districts of Izmir including tap water (n = 44), well water (n = 31), pool water (n = 26), lake water (n = 18), dam water (n = 10), stream (n = 9), seawater (n = 8) and thermal spring water (n = 2). From the total of samples, 71 (47.97%) were found to be positive for FLA. Eighteen out of 148 water samples (12.16%) included in this study were found positive for *Acanthamoeba* spp. according to Page's morphological analysis criteria (Fig 1). *Acanthamoeba* were detected in various water sources including 12.9% (4/31) WW, 44.44% (4/9) StW, 6.81% (3/44) TW, 11.53% (3/26) PW, 11.11% (2/18) LW, and 25% (2/8) SeW (Table 1).

### Pathogenic potential of *Acanthamoeba* spp. in positive water samples

The results of the tolerance assay of 18 *Acanthamoeba* isolates grown in culture were shown in Table 2. All of the 18 *Acanthamoeba* isolates investigated were grown at 37°C and 0.5 M mannitol. Ten (55.5%) of the isolates were grown both at 42°C, and 1 M mannitol. Only eight (44.4%) of the isolates were not grown at 42°C and 1 M mannitol. Therefore, ten isolates

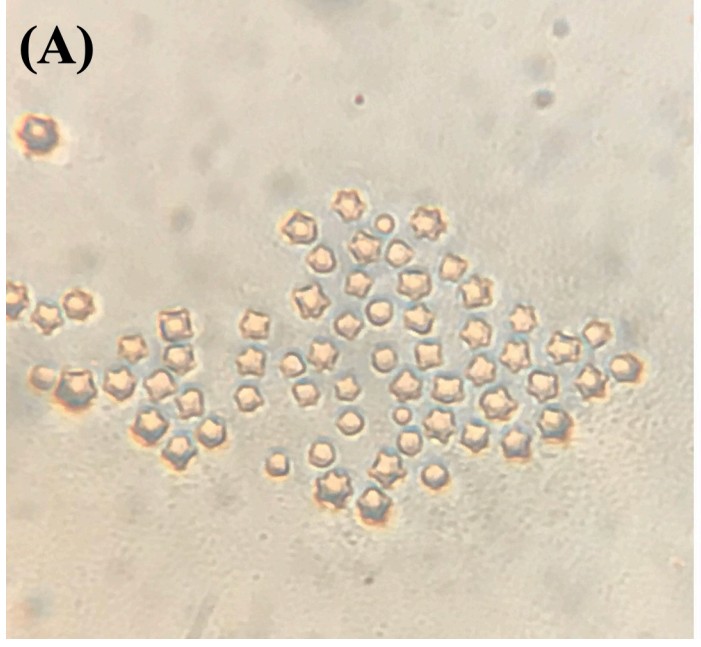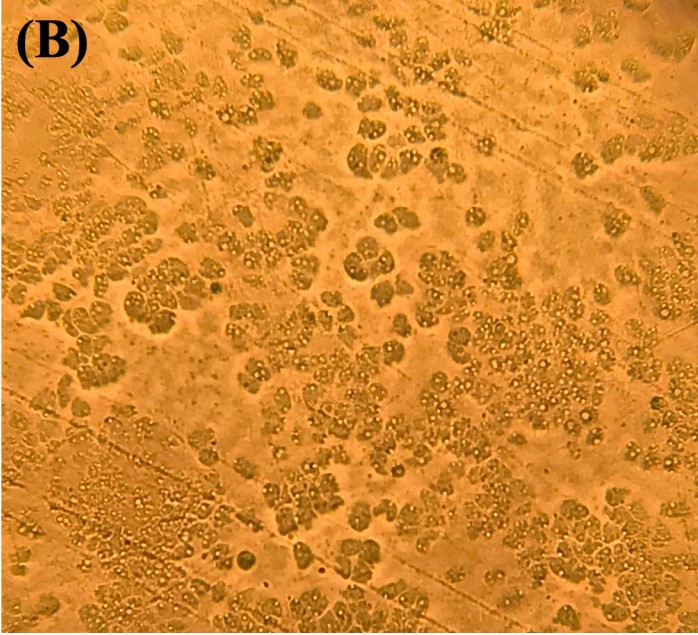

**Fig 1. *Acanthamoeba* cysts and trophozoites in non-nutrient agar plate found in various water sources collected from in İzmir, Turkey.** A) *Acanthamoeba* cysts, B) *Acanthamoeba* trophozoites.

**Table 1. Result of the qPCR assay and culture for *Acanthamoeba* spp. in various waters sources.**

| Water sources | Sample No. | Culture | | qPCR | | Mean Ct value | *Acanthamoeba* spp. (plasmid copies/l) |
|---|---|---|---|---|---|---|---|
| | | N | (%) | N | (%) | | |
| StW | 9 | 4 | 44.44 | 4 | 44.44 | 32.9 | $8.8 \times 10^4$-$7.3 \times 10^2$ |
| LW | 18 | 2 | 11.11 | 5 | 27.77 | 37.2 | $1.2 \times 10^3$-$1.4 \times 10^2$ |
| SeW | 8 | 2 | 25 | 2 | 25 | 31.2 | $3.2 \times 10^5$-$5.3 \times 10^3$ |
| PW | 26 | 3 | 11.53 | 6 | 23.07 | 36.8 | $5.3 \times 10^3$-$1.6 \times 10^2$ |
| WW | 31 | 4 | 12.9 | 6 | 19.35 | 34.4 | $1.6 \times 10^4$-$1.4 \times 10^3$ |
| TW | 44 | 3 | 6.81 | 4 | 9.09 | 36.1 | $1.9 \times 10^4$-$1.8 \times 10^2$ |
| **Total** | **136** | **18** | **13.23** | **27** | **19.85** | | |

**TW:** Tap water, **PW:** Pool water, **WW:** Well water, **LW:** Lake water, **StW:** Stream water, **SeW:** Seawater

(55.5%) were considered potentially pathogenic, and the rest (44.4%) were classified as low pathogenic potential.

The growth ability was evaluated by exposure at different temperatures (37˚-42˚C) and osmolarity ranges (0.5-1M mannitol) to test the pathogenicity potential of 18 *Acanthamoeba* positive samples in NNA plate culture. The isolates that can grow at high temperature (at 42˚C) and high osmolarity (1 M mannitol) were considered as potential pathogenic strains. However, the other isolates that were able to grow at 37˚C temperature and 0.5 M osmolarity were considered as low pathogens.

**Table 2. Thermo/osmo-tolerance assay of *Acanthamoeba* positive strains isolated from water samples in different districts of İzmir province.**

| Strain No. | Sampling area | Locality | NNA | qPCR | PCR (JDP) | Species | Genotype | Tolerance Assay | |
|---|---|---|---|---|---|---|---|---|---|
| | | | | | | | | Osmo-tolerance (M, mannitol) | Thermo-tolerance (˚C) |
| | | | | | | | | 0.5 / 1 | 37 / 42 |
| IWS3 | Tap water | Torbalı | + | + | + | *Acanthamoeba* sp. | T4 | +/+ | +/+ |
| IWS24 | Tap water | Torbalı | + | + | + | *Acanthamoeba* sp. | T4 | +/+ | +/+ |
| IWS30 | Tap water | Çeşme | + | + | + | *Acanthamoeba* sp. | T5 | +/+ | +/+ |
| IWS47 | Pool water | Bornova | + | + | + | *Acanthamoeba* sp. | T2 | +/- | +/- |
| IWS56 | Pool water | Bornova | + | + | + | *Acanthamoeba* sp. | T5 | +/- | +/- |
| IWS60 | Pool water | Bornova | + | + | + | *Acanthamoeba* sp. | T4 | +/+ | +/+ |
| IWS78 | Well water | Bayraklı | + | + | + | *Acanthamoeba* sp. | T4 | +/+ | +/+ |
| IWS80 | Well water | Dikili | + | + | + | *Acanthamoeba* sp. | T15 | +/- | +/- |
| IWS93 | Well water | Bayraklı | + | + | + | *Acanthamoeba* sp. | T4 | +/+ | +/+ |
| IWS95 | Well water | Dikili | + | + | + | *Acanthamoeba* sp. | T2 | +/+ | +/+ |
| IWS103 | Lake water | Menderes | + | + | + | *Acanthamoeba* sp. | T5 | +/+ | +/+ |
| IWS114 | Lake water | Ödemiş | + | + | + | *Acanthamoeba* sp. | T4 | +/- | +/- |
| IWS130 | Stream water | Menderes | + | + | + | *Acanthamoeba* sp. | T2 | +/- | +/- |
| IWS132 | Stream water | Menderes | + | + | + | *Acanthamoeba* sp. | T5 | +/+ | +/+ |
| IWS134 | Stream water | Menderes | + | + | + | *Acanthamoeba* sp. | T4 | +/+ | +/+ |
| IWS136 | Stream water | Menderes | + | + | + | *Acanthamoeba* sp. | T4 | +/- | +/- |
| IWS142 | Seawater | Çeşme | + | + | + | *Acanthamoeba* sp. | T15 | +/- | +/- |
| IWS146 | Seawater | Çeşme | + | + | + | *Acanthamoeba* sp. | T4 | +/- | +/- |

(+): Positive samples, (-): Negative samples

## PCR amplification efficiency (*E*) and standard curve

A standard curve was developed by serial dilutions of known amounts of plasmid DNA ranging $10^9$ to $10^0$ plasmids copy from *Acanthamoeba*, *N. fowleri*, and *B. mandrillaris*. The standard curve of the known plasmid concentrations and C*t* values of *Acanthamoeba*, *N. fowleri*, and *B. mandrillaris* was shown in Fig 2. A slope (*S*) of -3.37, -3.58, and -3.35 equals a PCR amplification efficiency (*E*) of 97.8%, 90.2% and 98.5% for *Acanthamoeba*, *N. fowleri* and *B. mandrillaris*, respectively. Moreover, the determination coefficient ($R^2$) of 0,99, 0,99, and 0,98 were observed for *Acanthamoeba*, *N. fowleri*, and *B. mandrillaris*, respectively. The lowest detection limit for *Acanthamoeba*, and *N. fowleri* of the qPCR assay was determined as one plasmid copy per reaction. However, the lowest detection limit for *B. mandrillaris* was set at 10 plasmid control DNA copies.

## Quantification of *Acanthamoeba* and *N. fowleri* from various water

A total of 27/148 (18.24%) *Acanthamoeba* spp. and 4/148 (2.7%) *N. fowleri* positives were detected in six different water sources by the qPCR assay. However, all water samples were found negative for *B. mandrillaris*. The qPCR assay was applied to confirm positive samples in terms of FLA in culture, and 18 (12.16%) *Acanthamoeba* and four (2.7%) *N. fowleri* were found to be positive. The overall presence of *Acanthamoeba* spp. in various water sources were 4/9 (44.44%) in StW, 5/18 (27.77%) in LW, 2/8 (25%) in SeW, 6/26 (23.07%) in PW, 6/31 (19.35%) in WW, and 4/44 (9.09%) in TW (Table 1).

*Acanthamoeba* spp. were found more frequent in streams samples than tap waters. However, it could not detect in dam water and thermal spring water. The C*t* value of PCR amplification of *Acanthamoeba* in StW, LW, SeW, PW, WW, and TW in the ranges from 28.2 to 39.4, and also plasmid copies concentrations of *Acanthamoeba* 18S rRNA gene were detected in the range of $3.2 \times 10^5$-$1.4 \times 10^2$ plasmid copies/l. In our study, the highest concentration of *Acanthamoeba* was found quantitatively in seawater samples, while the lowest concentration was found in lake water (Fig 3).

*N. fowleri* was detected in various water sources, 2/26 (7.69%) in PW, 1/18 (5.5%) in LW, and 1/10 (10%) in DW samples. For various water sources in PW, LW, and DW, means of C*t* values was ranged from 35.13 to 38.13, and plasmid copies of *N. fowleri* were between $8 \times 10^3$ and $11 \times 10^2$ plasmid copies/l. The average highest concentration of *N. fowleri* was shown in dam water (Table 3). Two water samples which sample code PW12 and LW4 were detected positive both *Acanthamoeba* and *N. fowleri*.

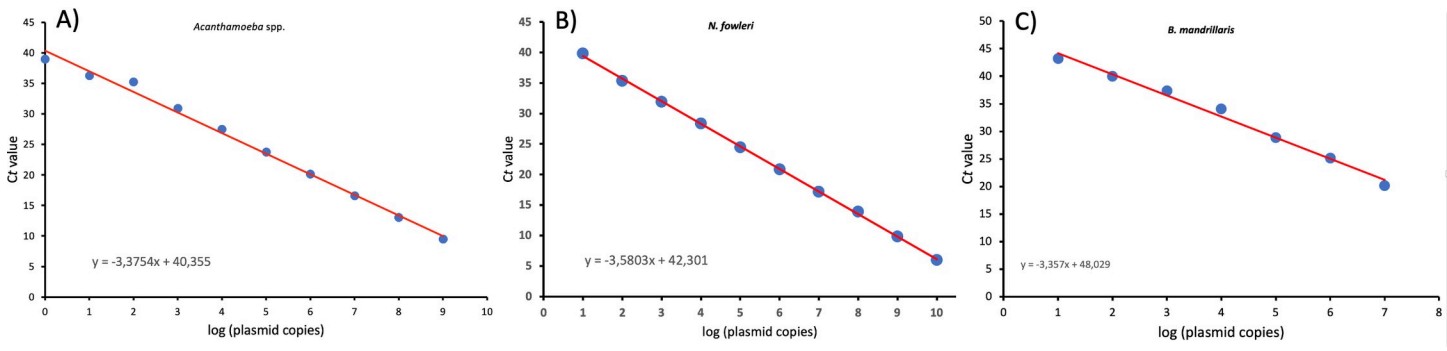

**Fig 2.** Standard curves were generated by linear regression of the cycle threshold (C*t*) versus (A) *Acanthamoeba*, (B) *N. fowleri*, and (C) *B. mandrillaris* plasmid control DNAs.

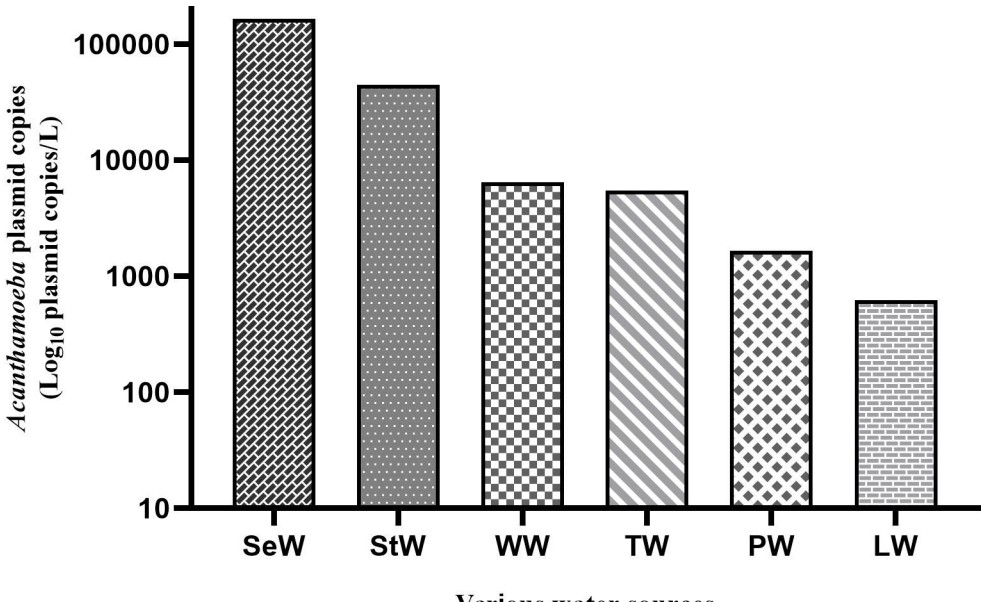

**Fig 3. DNA quantity of *Acanthamoeba* plasmid copies determined by qPCR assay in various water sources from İzmir, Turkey.**

## Genotyping of *Acanthamoeba* spp. and *N. fowleri*-positives in water samples

All 27 *Acanthamoeba* isolates detected positive by qPCR were subjected for PCR amplification using JDP primer sets, which is specific for the DF3 region of the 18S rRNA gene sequences. The sequence data obtained from *Acanthamoeba* isolates were aligned using Clustal W software and were used to construct the phylogenetic tree to illustrate the relationships between the isolates obtained and reference sequences of *Acanthamoeba* genotypes retrieved from Gen-Bank (*A. palestinensis* genotype T2 accession nos: U07411 and L09599; *A. castellanii* genotype T4 accession nos: MF806034, KT892904, MF139789,MH620482, MH620483, MK192795 and MG969963; *A. lenticulate* genotype T5 accession nos: KX018036, U94730, EU377584, U94740, U94737 and U94736; *A. jacobsi* genotype T15 accession nos:KX870203, KT892847, GQ905495 and MH790995; *Acanthamoeba* genotype T17 accession no:GU808277; *Acanthamoeba* genotype T18 accession no:KC822461). All samples showed nucleotide identity between 98% and 100% with reference strains deposited in GenBank (BLASTn) (www.ncbi.nlm.nih.gov/BLAST). In this study, four different genotypes (T2, T4, T5, and T15) were detected in water samples. Fourteen (51.85%) of 27 *Acanthamoeba* isolates were belonged to T4 genotype. The

**Table 3. Result of the qPCR assay for *Naegleria fowleri* in various waters.**

| Strain No. | Sample code | Mean Ct value | *N. fowleri* (plasmid copies/l) |
|---|---|---|---|
| IWS52 | PW8 | 38.13 | $11 \times 10^2$ |
| IWS56 | PW12 | 37.71 | $15 \times 10^2$ |
| IWS105 | LW4 | 36.68 | $3 \times 10^3$ |
| IWS120 | DW1 | 35.13 | $8 \times 10^3$ |

**IWS:** Izmir water sample, **PW:** Pool water, **LW:** Lake water, **DW:** Dam water

remaining isolates were belonging to T5 genotype 6/27 (22.22%), T2 genotype 4/27 (14.81%) and T15 genotype 3/27 (11.11%). The distribution of *Acanthamoeba* genotypes according to different districts of İzmir province and various water resources was showed in Fig 4 and Table 4. According to the phylogenetic tree, *Acanthamoeba* isolates T4 obtained from various water sources were grouped within the clade including the other sequences of *Acanthamoeba castellanii* complex available from Genbank. Six isolates (IWS_132, IWS_79, IWS_30, IWS_56, IWS_68, IWS_103) were found to be *Acanthamoeba* genotype T5 revealing 98% sequence identity to various T5 reference strain. Phylogenetic tree showed that three isolates (IWS_61, IWS_80 and IWS_142) were strictly related with *Acanthamoeba* T15 genotype chosen as references with 100% of identity, four isolates (IWS_14, IWS_47, IWS_95 and IWS_130) T2 genotype with 98% of identity with the T2 sequence references (Fig 5).

Four positive samples detected *N. fowleri* by qPCR were shown to be 99–100% nucleotide identity with references isolates in GenBank (*N. fowleri* accession nos: AJ132028, X96565, AJ132019 and X96564). Genotypic differences of *N. fowleri* can be distinguished based on a one bp transition in 5.8S rRNA and the length of the internally transcribed spacer (1) [23]. Among the *N. fowleri* positive water samples (DW1, LW1and PW12), three of them do have T at position 31 in the 5.8S rRNA, and the ITS1 length was 42 bp, so these were identified as belonging to type 2. However, in one *N. fowleri* (PW8) sample, the sequence C has not changed to T at position 31 in 5.8S rRNA and the ITS1 length was found to be 84 bp with a two bp deletion in the sequence of repeat. According to obtained data this sample was identified as type 5 (Table 5). The phylogenetic tree was shown using neighbor-joining models of 5.8S rRNA, ITS1, and ITS2 sequence data for *Naegleria* spp. (Fig 6). According to the phylogenetic tree, *N.*

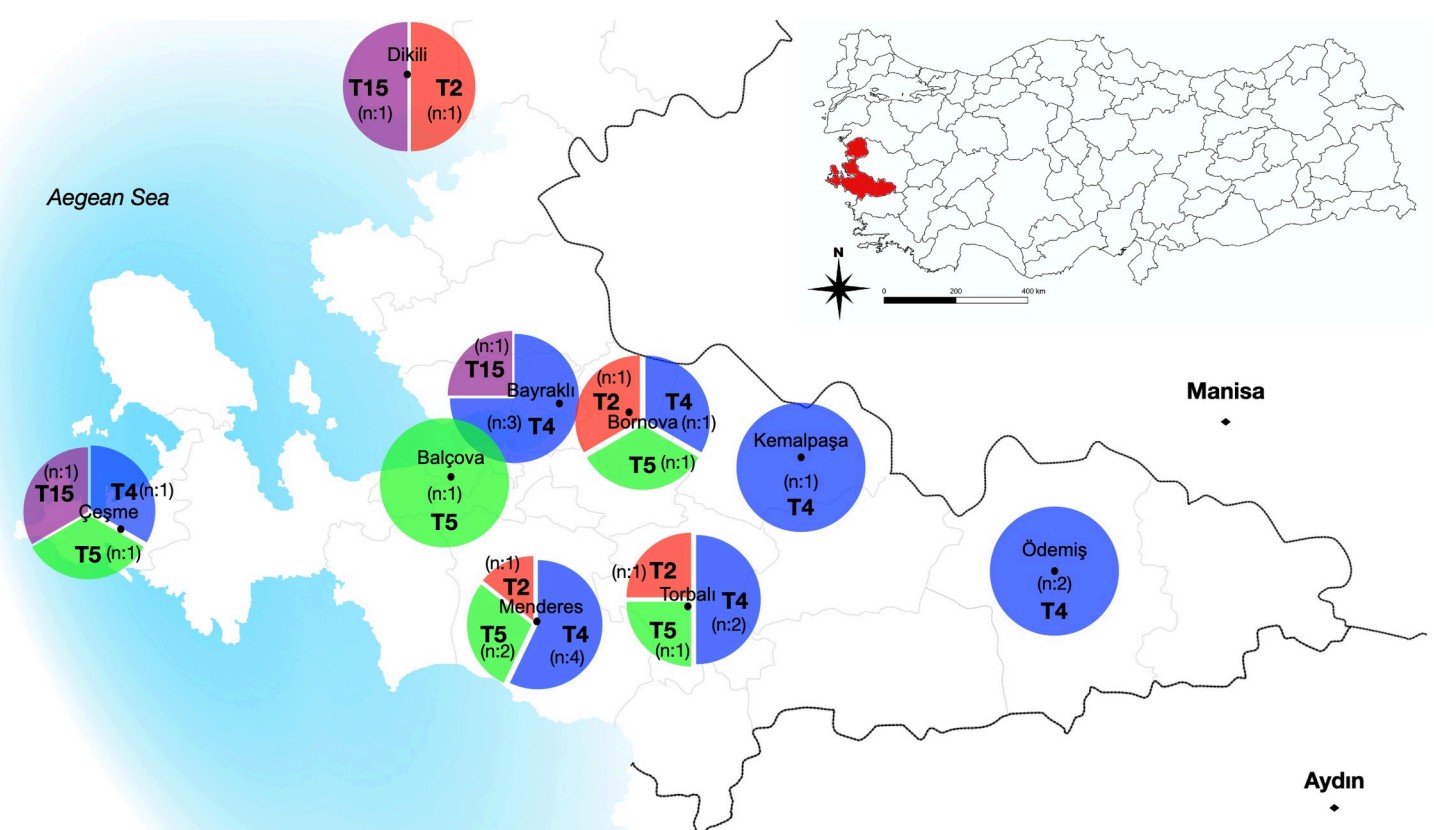

**Fig 4. The distribution of *Acanthamoeba* genotypes according to different districts of İzmir province and various water resources.**

**Table 4. Distribution of *Acanthamoeba* genotype in various water sources.**

| Genotype | Total | | LW | WW | TW | StW | PW | SeW |
|---|---|---|---|---|---|---|---|---|
| | % | n | | | | | | |
| T4 | 51.85 | 14 | 4 | 3 | 2 | 2 | 2 | 1 |
| T5 | 22.22 | 6 | 1 | 1 | 1 | 1 | 1 | |
| T2 | 14.81 | 4 | | 1 | 1 | 1 | 1 | |
| T15 | 11.11 | 3 | | 1 | | | 1 | 1 |
| Total | 100 | 27 | 5 | 6 | 4 | 4 | 5 | 2 |

**TW:** Tap water, **PW:** Pool water, **WW:** Well water, **LW:** Lake water, **StW:** Stream water, **SeW:** Seawater

*fowleri* isolates obtained from various water sources were grouped within the clade including the other sequences of *N. fowleri* genotypes available from GenBank. Three isolates (LW1, DW1 and PW12) were found to be *N. fowleri* type 2 revealing 100% sequence identity to various *N. fowleri* type 2 reference strains (*N. fowleri* type 2 accession nos: AJ132019 and X96564). However, phylogenetic tree showed that one isolates (PW8) were strictly related with *N. fowleri* type 5 chosen as references (*N. fowleri* type 5 accession nos: AJ132028 and X96565) with 99% of identity. In the present study, *N. fowleri* type 2 and type 5 were identified in water samples for the first in Turkey.

### Associations between *Acanthamoeba* and water quality parameter variables

A non-parametric test was conducted to determine the relationship between five different water quality parameters and the presence/absence of *Acanthamoeba* in various water sources. The result of the non-parametric statistical test was shown in Table 6. A significant relationship with Mann-Whitney *U* test was found between the presence/absence of *Acanthamoeba* spp. and pH values from well water and also EC-TDS values from pool water, lake water, and sea waters ($p < 0.05$).

### Discussion

Among the FLA, *Acanthamoeba* spp., *N. fowleri*, and *B. mandrillaris* are eukaryotic protists widely found in many places of the world. These FLAs can potentially cause opportunistic/non-opportunistic infections in humans and animals [1, 24]. Recently, they have received increasing attention in the medical and scientific world due to the serious fatal infections in humans. The impact of *Acanthamoeba* and *N. fowleri* on human health is associated with the genotypes of these pathogens and their reproduction in water and soil resources, a natural reservoir location. Therefore, preventive, and investigational monitoring programs for measuring the density of *Acanthamoeba* and *N. fowleri* are important in aquatic environments with human exposure, which can be achieved with real-time qPCR [3]. This study was occurred the quantify presence of *Acanthamoeba* and *N. fowleri* from various water sources in province İzmir, Turkey. *Acanthamoeba* spp. were found positive ranging from 4.4% to 50% in various water sources such as tap water, ponds, rivers, streams, and water wells in Turkey [25–29]. *Acanthamoeba* was detected positively 17.3%, 28.8%, 15.9% and 42.9% in Jamaica, Iran, Thailand, and Uganda in tap water sources, respectively [30–33]. The results of this study which indicate that the *Acanthamoeba* spp. occurrence in various water sources (18.24%) are similar to those results obtained in the America, Brazil, and Japan Jamaica, Iran, Thailand, Turkey. In previous reports, *Naegleria* spp. was found in various water sources worldwide at 0.6%–60.9% all over the world, and 0.7–10% in Turkey. [26, 34, 35]. We conclude that *Naegleria* spp. and

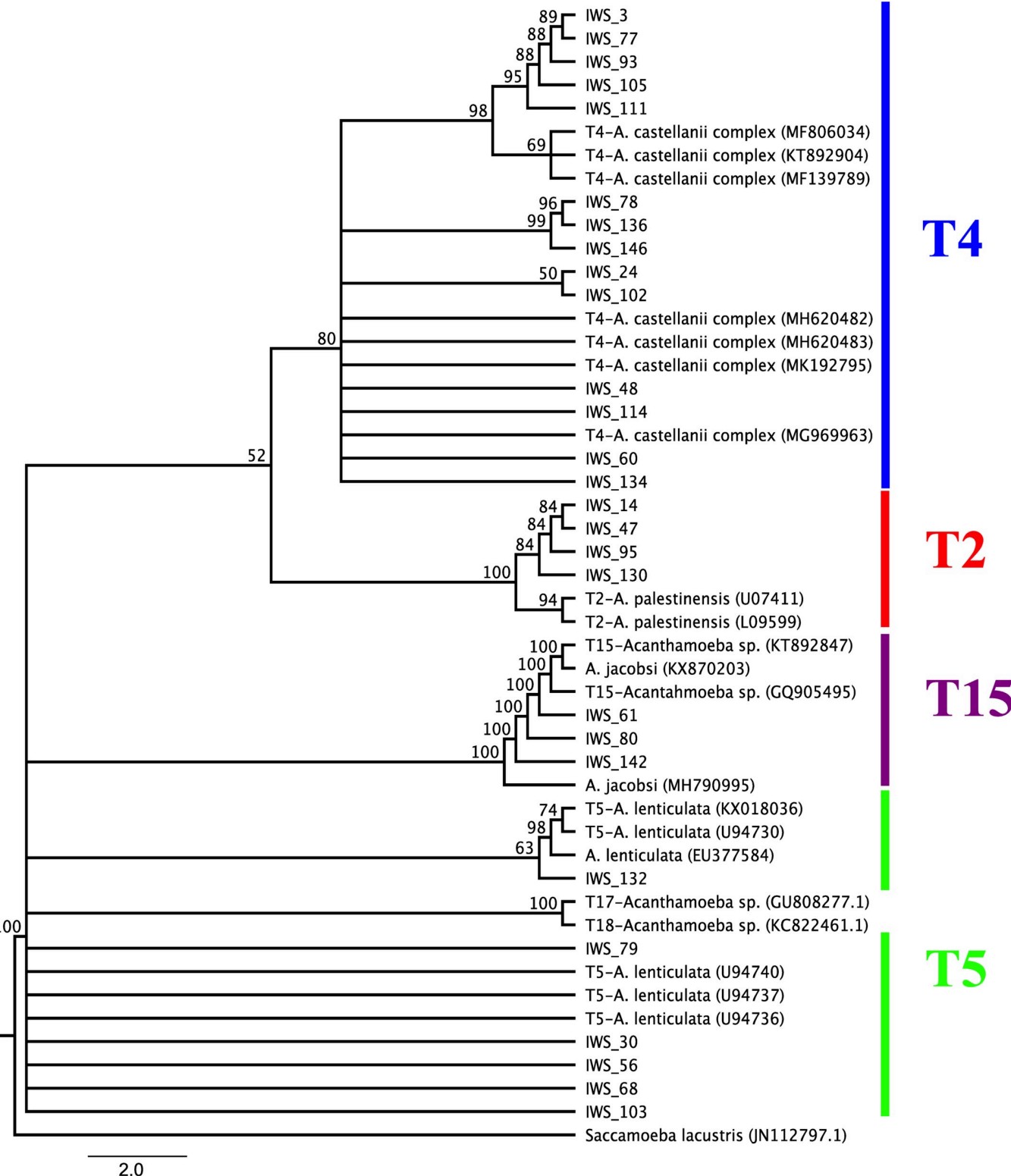

**Fig 5. Phylogenetic tree inferred using neighbor-joining models of the 18S rRNA gene DF3 region sequence data for the *Acanthamoeba* genotypes by MEGA X.** Bootstrap values are based on 1000 replicates and the root of the tree evaluated by an outgroup *Saccamoeba lacustris*.

**Table 5. Base length and position of ITS1, ITS2 and 5.8S rRNA sequences of references strains and all the positive isolates in this study.**

| Isolate | Genotype | ITS1 | 5.8S | Position 31 in 5.8S | ITS2 | Total length | Accession number | Reference |
|---------|----------|------|------|---------------------|------|--------------|------------------|-----------|
| 7853 | T1 | 42 | 175 | C | 106 | 323 | AY376149 | [1] |
| AR12 | T2 | 42 | 175 | T | 106 | 323 | X96564 | [2] |
| LEE | T3 | 86 | 175 | T | 106 | 367 | X96562 | [2] |
| Ch2-1-f2 | T4 | 86 | 175 | T | 106 | 367 | AJ132030 | [3] |
| Na 420c | T5 | 84 | 175 | C | 106 | 365 | AJ132028 | [3] |
| J2B2 | T6 | 114 | 175 | C | 106 | 395 | FR875287 | [4] |
| M4E | T7 | 142 | 175 | T | 106 | 423 | X96563 | [2] |
| C0504 | T8 | 130 | 175 | T | 106 | 411 | FR875288 | [4] |
| DW1 | T2 | 42 | 175 | T | 106 | 323 | MW677629 | This study |
| LW1 | T2 | 42 | 175 | T | 106 | 323 | MW677627 | This study |
| PW12 | T2 | 42 | 175 | T | 106 | 323 | MW676178 | This study |
| PW8 | T5 | 84 | 175 | C | 106 | 365 | MW677628 | This study |

[1] Zhou et al., 2003.

[2] De Jonckheere, 1998.

[3] Pélandakis et al., 2000.

[4] De Jonckheere, unpublished.

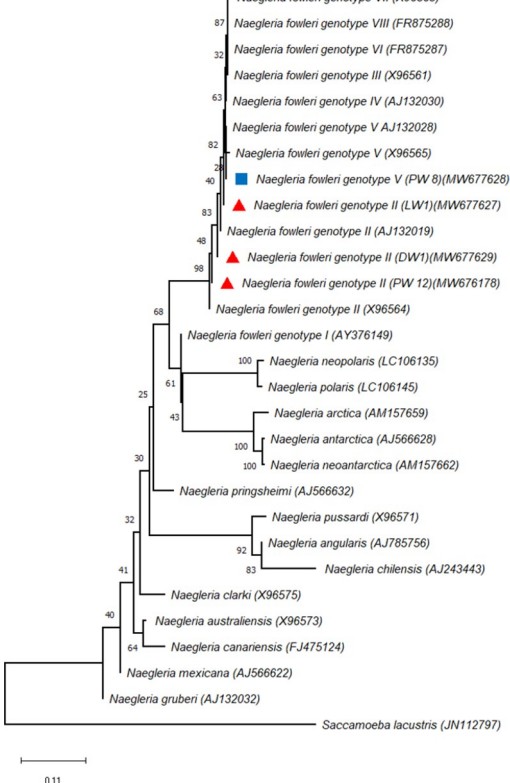

**Fig 6. Phylogenetic tree of *Naegleria* isolates for the ITS sequences.** Phylogenetic tree inferred using neighbor-joining cluster analysis of the sequence obtained and the sequences from ITS1, 5.8S rRNA and ITS2 sequences of various *Naegleria* strains, with special reference to *N. fowleri* produced in MEGA X. Bootstrap values are based on 1000 replicates.

**Table 6. Result of non-parametric test showing the relationship between five different water quality parameters and the presence/absence of *Acanthamoeba* spp. in various water sources.**

| Water quality parameters | *Mann-Whitney U* test | | | | | |
|---|---|---|---|---|---|---|
| | **TW** | **PW** | **WW** | **LW** | **StW** | **SeW** |
| pH value | P = 0,390 | P = 0,223 | **P = 0,004***  | P = 0,692 | P = 0,135 | P = 0,495 |
| Temperature (˚C) | P = 0,744 | P = 0,271 | P = 0,515 | P = 0,235 | P = 0,618 | P = 0,495 |
| Free Cl (ppm (mg/l)) | P = 0,163 | P = 1,000 | P = 0,118 | P = 1,000 | P = 0,091 | P = 1,000 |
| [a] EC (0–9990 μs/cm) | P = 0,624 | **P = 0,002*** | P = 0,146 | **P = 0,002*** | P = 0,135 | **P = 0,008*** |
| [b] TDS (0–9990 ppm) | P = 0,540 | **P = 0,002*** | P = 0,051 | **P = 0,002*** | P = 0,618 | **P = 0,040*** |

[a] Electrical conductivity,

[b] Total dissolved solids

* Statistically significant $p < 0.05$

**TW:** Tap water, **PW:** Pool water, **WW:** Well water, **LW:** Lake water, **StW:** Stream water, **SeW:** Seawater

*Acanthamoeba* spp. are free-living amoebas that have suitable growth in various water sources worldwide, but detection rates at different regions may be influenced by water types and geographical conditions.

The qPCR assay is a method that can effectively detect and quantify amoeba even it was failed to culture due to low sample. However, standard curves characterizing the relationship between plasmid copy number and qPCR data can facilitate the quantification of microorganisms in environmental water samples [8, 17]. In this study, the lowest detection limit (one plasmid copy) was detected for *Acanthamoeba* and *N. fowleri* for each reaction by qPCR method. In previous reports, similar results was identified the lowest detection limit of one plasmid copy DNA [9]. In this study, the quantitative amount of *Acanthamoeba* spp. DNA obtained from direct water samples was ranged from $3.2 \times 10^5$ to $1.4 \times 10^2$ plasmid copies/l. In previous studies performed in Germany and Taiwan, *Acanthamoeba* spp. was reported to range from $2.0–3.0 \times 10^3$ and $2.0 \times 10^2–9.0 \times 10^4$ and $3.4–5.0 \times 10^3–2.2–1.4 \times 10^3$ amoebae/l, respectively [8, 36]. We thought that might be the reason why *Acanthamoeba* is detected in high concentrations in seawater samples is due to the mixing of sewage wastes or industrial wastewater to the beaches. Moreover, The World Health Organization (WHO 2006) reported the presence of *Acanthamoeba* in seawater to be associated with sewage and waste effluent outlets. Lorenzo-Morales et al. reported the high rate of isolation (49.6% and 64.0%) for *Acanthamoeba* in seawater due to the sewage-waste and industrial effluent [30, 37]. In our study, the quantitative amount of *N. fowleri* in dam, lake, and pool water samples was determined as between $11 \times 10^2$ and $8.0 \times 10^3$ plasmid copies/l. Recent study performed in Belgium was reported that *Naegleria* spp. in cooling water samples was $6.3 \times 102–4.1 \times 10^3$ cells/l [7]. In Australia, *Naegleria* spp. have been reported in the range of $4–3.4 \times 10^2$ cells/l in the drinking water sample [38]. *Naegleria* spp. concentration has been reported in the range of 1.1–24.2 cells/l in hot spring and drinking water samples in Taiwan [39].

Up to date, 22 genotypes (T1-22) of *Acanthamoeba* have been identified as a result of the sequence analysis of the DF3 region of the 18S rRNA gene, but this classification containing both pathogens and non-pathogens genotypes [3, 40]. Among these genotypes, it has been reported that the T4 genotype was found the most common in environmental and clinical samples and is the pathogen of different diseases (AK, GAE, skin lesions) [5, 41]. Furthermore, the other genotypes including T2, T3, T5, T6, T10, T11, T12, T15, and T18 was reported the association with human infections [1, 42, 43]. In our study, genotype T4 (51.85%) was determined as the most common genotype in water samples. Apart from that, T5 (22.22%), T2

(14.81%), and T15 (11.11%) genotypes were detected in water samples. In earlier studies, genotypes T1, T2, T3, T4, and T7 of *Acanthamoeba* spp. were detected in freshwater resources in Egypt [44]. In another study, the genotype T4 (51%), as well as T14 (18%), T5 (11%), T3, T15, T16 and T10 (4% per each), T11 (3%) and T7, T9 (1% per each) was reported the most common genotype in water samples in Tunisia [45]. The most common genotypes, T4 (93.7%) and T2 (6.25%) were determined in the geothermal river in the southwest of Iran by Niyyati et al. [46]. The most common T4 and T2, T3, T5, T11, T15 genotypes of *Acanthamoeba* were detected in environmental water samples in Turkey [34, 47–50]. In the previous study T4 and T5 genotypes were reported in keratitis wild birds in İzmir [51]. Moreover, *Acanthamoeba* T4 genotypes were detected in corneal scraping samples taken from patients with suspected *Acanthamoeba* keratitis in Turkey [52, 53]. *Acanthamoeba* genotypes T4 and T5 commonly found in environmental samples pose a greater risk to humans and animals.

*N. fowleri*, which is the pathogen type of the *Naegleria* genus for humans, causes fatal PAM. *N. fowleri* is also a protist pathogen widely found in the environment including rivers, lakes, dams, hot springs, geothermal springs, untreated and treated domestic water sources, and swimming pools [54, 55]. In this study, *N. fowleri* was found positive for the first time in environmental water resources collected in İzmir province, Turkey. *N. fowleri* was found positively in various water sources including in 7.69% (2/26) pool water, 10% (1/10) dam water and 5.5% (1/18) lake water. The presence of *Naegleria* species has been reported to be positively detected in environmental water samples at a rate of 13.2–60.9% in Europe, 3–46% in the USA and 0.6–56.9% in Asia [7, 56–58]. Up to date, there are 47 species in the genus of *Naegleria* and also eight genotype of *N. fowleri* [58, 59] were identified. *N. fowleri* genotypes are characterized that difference 1, 2, 3, and 4 types are the only C to T transition at position 31 in the 5.8S rRNA sequence, because of the equal of the ITS lengths. However, *N. fowleri* type 5 has not identical the ITS1 lengths with the T at position 31 in the 5.8S rRNA sequence [23, 60, 61]. In this study *N. fowleri* type 2 and type 5 were isolated for the first time from water sources in Turkey. *N. fowleri* type 2 and 3 have been reported in many patients and water samples worldwide. Type 2, 3, 4, 5, 6, 7, and 8 in Europe, types 1, 2, and 3 in the USA, types 2 and 3 in Asia and only one type of type 5 in the Western Pacific (Oceania and Japan) has been found in human specimens and water samples [18, 23, 62]. Since there are a limited number of studies typing *N. fowleri*, the information about the pathogenicity of the types is insufficient. However, there is not yet conclusive evidence of any difference in virulence for any of the detected *N. fowleri* types. It is likely to be detected in humans, as types 2, 3, and 5 are the most common in waters.

## Conclusions

In conclusion, the present study reports both the presence and the concentration of *Acanthamoeba* and *N. fowleri* in various water sources and demonstrates their rapidly determination by qPCR. Although the presence of *Acanthamoeba* was found higher in stream samples, the quantitative value of *Acanthamoeba* was detected higher in seawater samples. Therefore, it should be kept in mind that it may pose a risk for people during sea activities in summer in Izmir, which is a holiday region. *Acanthamoeba* T4 and T5 genotypes, which are commonly detected as causal agents of AK and GAE infection, were found at a high rate in various water sources in our study. In this study, *N. fowleri* type 2 and type 5 were isolated the first time from water sources in Turkey. Since the genotypes of *Acanthamoeba* spp. and *N. fowleri* types can be detected in many environments, they have pathogenic potentials that may pose a risk to human health. For this reason, a mandatory inspection is necessary, especially in potable waters, as it may pose a risk to people in swimming and recreational waters. In this study, the presence of pathogenic potential of the identified strains has been revealed. However, further

studies needs to be done using environmental samples across Turkey. Also, clinicians and public health professionals should increase awareness about these issues by the help of civilian authorities.

## Supporting information

**S1 Table. Various water samples collected from İzmir region and their geographic coordination.**
(DOCX)

**S2 Table. The set of primers and probes in 18S rRNA and ITS gene amplification for *Acanthamoeba* spp., *B. mandrillaris* and *N. fowleri*.**
(DOCX)

## Acknowledgments

We are grateful to Ibne Karim M. Ali who provided *N. fowleri* and *B. mandrillaris* positive control DNA from Free-Living and Intestinal Amebas (FLIA) Lab, CDC, Atlanta, Georgia, USA. We would like to thank the colleague Dr. Mehmet Karakus for editing an earlier draft of this manuscript.

## Author Contributions

**Conceptualization:** Mehmet Aykur, Hande Dagci.

**Data curation:** Mehmet Aykur.

**Formal analysis:** Mehmet Aykur.

**Investigation:** Mehmet Aykur.

**Methodology:** Mehmet Aykur.

**Project administration:** Hande Dagci.

**Resources:** Hande Dagci.

**Supervision:** Hande Dagci.

**Visualization:** Mehmet Aykur.

**Writing – original draft:** Mehmet Aykur.

**Writing – review & editing:** Hande Dagci.

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
