## [Decision Letter · Decision Letter 0]

14 Jun 2021

PONE-D-21-09475

Evaluation of molecular characterization and phylogeny for quantification of Acanthamoeba and Naegleria fowleri in various water sources, Turkey

PLOS ONE

Dear Dr. AYKUR,

Thank you for submitting your manuscript to PLOS ONE. After careful consideration, we feel that it has merit but does not fully meet PLOS ONE’s publication criteria as it currently stands. Therefore, we invite you to submit a revised version of the manuscript that addresses the points raised during the review process.

We look forward to receiving your revised manuscript.

Kind regards,

Maria Stefania Latrofa

Academic Editor

PLOS ONE

Journal Requirements:

"The research was supported by a grant from the Budget of the Academic Staff Training Program by The Council of Higher Education and the Scientific Research Projects Branch Directorate of Ege University, Turkey (Project No: 18-TIP-025)."

 "The author(s) received no specific

funding for this work. The funders had no role in study design, data collection and

analysis, decision to publish, or preparation of the manuscript."

Additional Editor Comments:

Line 43: It is not clear which ITS region was amplified by cPCR. only ITS1 and 5.8S rRNA were mentioned; in the text it was  described that also ITS2 was also analysed. See lines 186 and 371.

Lines 73- 76: change, for example, into “Quantitative real-time PCR (qPCR) assay is a method with high specificity and sensitivity useful for detecting the presence of the amoebae in water resources (8,9).

Lines 77-83: delete the sentences or summarize this part being the usefulness of the qPCR is well known.

Lines 86- 88: delete the  sentence; it belongs to M&M section

Line 167: please check the reference (i.e., 17)

Lines 181-182: delete the sentence “In this study, positive …. and *B.*
*mandrillaris*.”

Line 186: specify “ITS”, the authors refer to ITS1, or ITS2 and 5.8S; see line 43

Line 273: change into “Table 1”, being the first table mentioned in the text

Line 281: change into “Table 2”

Lines 273 and 281: check the order of tables 1 and 2 throughout the text

Line 304: delete “in water samples collected from different districts of Izmir”

Line 334: delete “27”

Lines 335 and 362: change “homology” into “nucleotide identity”

Lines 337 and 363: add the accession number of the reference strains

Lines 341-344: phylogenetic analyses are poorly described; The phylogeny does not show homology among sequence but their phylogenetic relationship. Please modify the sentence accordingly. In the results only a comment about T4, what about the other genotypes detected?; *Acanthamoeba castellanii* has not been indicated in the tree; modify the figure adding for example “T4 *A. castellanii* complex”. I would suggest changing the sentence for example as: “According to the phylogenetic tree, *Acanthamoeba *isolates T4 obtained from various water sources were grouped within the clade including the other sequences of *Acanthamoeba castellanii *complex available from Genbank”

Lines 370- 372: the phylogenetic relationship of *N. fowleri* is not well define and described; the authors have to explain why the strain DW1 is included in a different clade respective to the other sequences having the genotype T2*.* The same is for the strain LW1. I suggest to use in the Mega X software, the best fit model program for the selection of the analysis model; I would suggest including an outgroup for both pathogens in the phylogenetic analyses.

Line 352-361: check this part. It could be included in another paragraph.

Change 5.8S rDNA into “5.8S rRNA”  throughout the ms.

The discussion needs to be improved by not including the results, but simply commenting on them, see for example lines 461-465.

Line 410: delete “by qPCR”

Figure:

Delete fig. 1, being superfluous.

Table

Table 2: please add the average of the CT value for *Acanthamoeba*

Table S2: specify which ITS region has been amplified in qPCR.

Reviewers' comments:

Reviewer's Responses to Questions

**Comments to the Author**

1. Is the manuscript technically sound, and do the data support the conclusions?

Reviewer #1: Yes

2. Has the statistical analysis been performed appropriately and rigorously? 

Reviewer #1: Yes

3. Have the authors made all data underlying the findings in their manuscript fully available?

Reviewer #1: Yes

4. Is the manuscript presented in an intelligible fashion and written in standard English?

Reviewer #1: No

5. Review Comments to the Author

Reviewer #1: Abstract

Line 44-45: "and that was the first-time detection in water sources, Turkey", reformulate this sentence

Line 49: To "the" best of our knowledge...

Introduction

Line 57: "are unicellular protozoa that find commonly in soil and water" commonly find

Line 67: "in particularly children and young adults" inmunocompetents childern and young adults

Material and methods

Lines 110-112: reformulate this sentence

Line 115: "It was filtrate in 0.22 μm pore-..."; write "through" instead of "in"

Line 130: Are the bacteria (E. coli) inactivated?

Line 136: "from" other organisms

Line 137: The grown "of" Acanthamoeba...was diferenciated from / Acanthamoeba trophs and cysts were diferenciated from

Osmo-tolerance assay: have you axenified the Acanthamoeba samples? were they in liquid culture? Why did you add E. coli in the tolerance assay plates? The tolerance asssay could not be trustable if the authors use a bacteria suspension

Line 181: Do you know the Acanthamoeba reference strain specie?

Line 184: Center(without s) for Disease Control and Prevention

Line 187: "quantification"

Results

Line 268: From the total of samples / From the 148 samples

Lines 282/283: ...were grown...

Table 1: positive / negative SAMPLES

As you could detect N.fowleri by qPCR, why you could not isolated it by NNA culture?

Discussion

Line 396: Reformulate this sentence

Lines398-406: the authors are presenting the results again. It is necessary to develope this paragraph by a comparison with other authors and reason the obtained results.

Line 453: including rivers...

Lines 455-457: Reformulate this sentence

It could be interesting if the authors talk about the pathogenicity of the different N.fowleri genotypes.

Conclusions

Lines 474-475: ...,THE present study reports both the presence and THE concentration" or "presence and concentration"..."and DEMONSTRATES THEIR RAPIDILY DETERMINATION by qPCR."

Line 479: "...which are commonly detected AS CAUSAL AGENTS OF AK..."

Line 488: "...should increase..."

6. PLOS authors have the option to publish the peer review history of their article (what does this mean?). If published, this will include your full peer review and any attached files.

Reviewer #1: No

---

## [Author Response · Author response to Decision Letter 0]

7 Jul 2021

Authors’ Responses to the Review Comments:

Editorial Office: Please ensure that your manuscript meets PLOS ONE's style requirements, including those for file naming. The PLOS ONE style templates can be found athttps://journals.plos.org/plosone/s/file?id=wjVg/PLOSOne_formatting_sample_main_body.pdf and https://journals.plos.org/plosone/s/file?id=ba62/PLOSOne_formatting_sample_title_authors_affiliations.pdf.

Response to Editor Office: As Editorial Office suggested, the manuscript was arranged in the PLOS ONE's style requirements

Editorial Office: Thank you for stating the following in the Acknowledgments Section of your manuscript:

"The research was supported by a grant from the Budget of the Academic Staff Training Program by The Council of Higher

Education and the Scientific Research Projects Branch Directorate of Ege University, Turkey (Project No: 18-TIP-025)."

We note that you have provided funding information that is not currently declared in your Funding Statement. However,

funding information should not appear in the Acknowledgments section or other areas of your manuscript. We will only

publish funding information present in the Funding Statement section of the online submission form.

Please remove any funding-related text from the manuscript and let us know how you would like to update your Funding

Statement. Currently, your Funding Statement reads as follows:

"The author(s) received no specific funding for this work. The funders had no role in study design, data collection and

analysis, decision to publish, or preparation of the manuscript."

Response to Editor Office: As Editorial Office suggested, all funding-related text from the manuscript have been removed from the manuscript. Additionally, the amended Finance Statement has been attached to the cover letter.

Additional Editor Comments:

Editorial Office: Line 43: It is not clear which ITS region was amplified by cPCR. only ITS1 and 5.8S rRNA were mentioned; in the text it was described that also ITS2 was also analysed. See lines 186 and 371.

Response to Editor Office: As Editorial Office suggested, line 41, 47 and line 192-193 it explained which ITS region was amplified by qPCR. The statement was added to article as follows.

Line 41: "N. fowleri positives were detected in six different water sources using qPCR which 5.8S rRNA gene and ITS regions (ITS1 and ITS2) specific primers". 

Line 47:"The four positive N. fowleri isolate was confirmed by sequencing the ITS1, ITS2 and 5.8S rRNA regions using specific primers"

Line 211-212: "the 5.8S rRNA and ITS (ITS1 and ITS2) regions for N. fowleri were selected".

Editorial Office: Lines 73- 76: change, for example, into “Quantitative real-time PCR (qPCR) assay is a method with high specificity and sensitivity useful for detecting the presence of the amoebae in water resources (8,9).

Response to Editor Office: As Editorial Office suggested, the statement in line 82-83 was changed as “Quantitative real-time PCR (qPCR) assay is a method with high specificity and sensitivity useful for detecting the presence of the amoebae in water resources (8,9)”.

Editorial Office: Lines 77-83: delete the sentences or summarize this part being the usefulness of the qPCR is well known.

Response to Editor Office: As Editorial Office suggested, the paragraph on line 77-83 has deleted.

Editorial Office: Lines 86- 88: delete the sentence; it belongs to M&M section

Response to Editor Office: As Editorial Office suggested, the sentence on line 86-88 has deleted.

Editorial Office: Line 167: please check the reference (i.e., 17)

Response to Editor Office: As Editorial Office suggested, the reference on line 193 (i.e., 17) has checked and removed. 

Editorial Office: Lines 181-182: delete the sentence “In this study, positive …. and B. mandrillaris.”

Response to Editor Office: As Editorial Office suggested, the sentence on line 207-208 has deleted.

Editorial Office: Line 186: specify “ITS”, the authors refer to ITS1, or ITS2 and 5.8S; see line 43

Response to Editor Office: As Editorial Office suggested, on line 211-212 "ITS" was indicated as ""the 5.8S rRNA and ITS (ITS1 and ITS2) regions for ". Also, references 15,16 and 17 were added.

Editorial Office: Line 273: change into “Table 1”, being the first table mentioned in the text

Response to Editor Office: As Editorial Office suggested, “Table 2” on line 306 was changed as “Table 1”

Editorial Office: Line 281: change into “Table 2”

Response to Editor Office: As Editorial Office suggested, “Table 1” on line 321 was changed as “Table 2”

Editorial Office: Lines 273 and 281: check the order of tables 1 and 2 throughout the text

Response to Editor Office: As Editorial Office suggested, on line 306 and line 321 checked the order of tables 1 and 2 throughout the text.

Editorial Office: Line 304: delete “in water samples collected from different districts of Izmir”

Response to Editor Office: As Editorial Office suggested, on line 361 the sentence “in water samples collected from different districts of Izmir” was deleted.

Editorial Office: Line 334: delete “27”

Response to Editor Office: As Editorial Office suggested, on line 403 “27” was deleted.

Editorial Office: Lines 335 and 362: change “homology” into “nucleotide identity”

Response to Editor Office: As Editorial Office suggested, on line 412 and 440 ““homology” was changed as “nucleotide identity”.

Editorial Office: Lines 337 and 363: add the accession number of the reference strains

Response to Editor Office: As Editorial Office suggested, on line 337 and 363 was added to article as follows the accession number of the reference strains.

Line 405-411: added the sentence “reference sequences of Acanthamoeba genotypes retrieved from GenBank (A. palestinensis genotype T2 accession nos: U07411 and L09599; A. castellanii genotype T4 accession nos: MF806034, KT892904, MF139789,MH620482, MH620483, MK192795 and MG969963; A. lenticulate genotype T5 accession nos: KX018036, U94730, EU377584, U94740, U94737 and U94736; A. jacobsi genotype T15 accession nos:KX870203, KT892847, GQ905495 and MH790995; Acanthamoeba genotype T17 accession no:GU808277; Acanthamoeba genotype T18 accession no:KC822461)”.

Line 440-441: added the sentence “references isolate in GenBank (N. fowleri accession nos: AJ132028, X96565, AJ132019 and X96564)”

Editorial Office: Lines 341-344: phylogenetic analyses are poorly described; The phylogeny does not show homology among sequence but their phylogenetic relationship. Please modify the sentence accordingly. In the results only a comment about T4, what about the other genotypes detected?; Acanthamoeba castellanii has not been indicated in the tree; modify the figure

adding for example “T4 A. castellanii complex”. I would suggest changing the sentence for example as: “According to the phylogenetic tree, Acanthamoeba isolates T4 obtained from various water sources were grouped within the clade including the other sequences of Acanthamoeba castellanii complex available from Genbank”

Response to Editor Office: As Editorial Office suggested, phylogenetic analysis was explained in detail and phylogenetic relationships between sequences were demonstrated. Also added comments about other genotypes and to article as follows.

Line 404-406: added the sentence “The sequence data obtained from Acanthamoeba isolates were aligned using Clustal W software and were used to construct the phylogenetic tree to illustrate the relationships between the isolates obtained and reference sequences of Acanthamoeba genotypes retrieved from GenBank”. 

Line 420-431: added the sentence “Six isolates (IWS_132, IWS_79, IWS_30, IWS_56, IWS_68, IWS_103) were found to be Acanthamoeba genotype T5 revealing 98% sequence identity to various T5 reference strain. Phylogenetic tree showed that three isolates (IWS_61, IWS_80 and IWS_142) were strictly related with Acanthamoeba T15 genotype chosen as references with 100% of identity, four isolates (IWS_14, IWS_47, IWS_95 and IWS_130) T2 genotype with 98% of identity with the T2 sequence references.”

As Editorial Office suggested, the figure of the phylogenetic tree was changed to “T4 A. castellanii complex”. Also, the sentence was changed as you suggested “According to the phylogenetic tree, Acanthamoeba isolates T4 obtained from various water sources were grouped within the clade including the other sequences of Acanthamoeba castellanii complex available from Genbank”.

Editorial Office: Lines 370- 372: the phylogenetic relationship of N. fowleri is not well define and described; the authors have to explain why the strain DW1 is included in a different clade respective to the other sequences having the genotype T2. The same is for the strain LW1. I suggest to use in the Mega X software, the best fit model program for the selection of the analysis model; I would suggest including an outgroup for both pathogens in the phylogenetic analyses.

Response to Editor Office: As Editorial Office suggested, phylogenetic analysis was explained in detail and phylogenetic relationships between sequences were demonstrated. Also added comments about other genotypes and to article as follows.

Line 475-482: added the sentence “According to the phylogenetic tree, N. fowleri isolates obtained from various water sources were grouped within the clade including the other sequences of N. fowleri genotypes available from GenBank. Three isolates (LW1, DW1 and PW12) were found to be N. fowleri type 2 revealing 98% sequence identity to various N. fowleri type 2 reference strains (N. fowleri type 2 accession nos: AJ132019 and X96564). However, phylogenetic tree showed that one isolates (PW8) were strictly related with N. fowleri type 5 chosen as references (N. fowleri type 5 accession nos: AJ132028 and X96565) with 99% of identity.” 

Moreover, As Editorial Office suggested, the phylogenetic tree was reconstructed using the Mega X software program and the optimal model, and an outgroup was added for the phylogenetic tree (Figure 7).

Editorial Office: Line 352-361: check this part. It could be included in another paragraph. Change 5.8S rDNA into “5.8S rRNA” throughout the ms.

Response to Editor Office: As Editorial Office suggested, the paragraph on lines 352-361 has been moved to lines 326-335. Also, "5.8S rDNA" in the entire manuscript was changed to "5.8S rRNA".

Editorial Office: The discussion needs to be improved by not including the results, but simply commenting on them, see for example lines 461-465.

Response to Editor Office: As Editorial Office suggested, it has been developed by making comments where necessary in the discussion section.

Editorial Office: Line 410: delete “by qPCR”

Response to Editor Office: As Editorial Office suggested, “by qPCR” on line 547 was deleted.

Editorial Office: Figure: Delete fig. 1, being superfluous.

Response to Editor Office: As Editorial Office suggested, figure 1 has been removed.

Editorial Office: Table 2: please add the average of the CT value for Acanthamoeba

Response to Editor Office: As Editorial Office suggested, table 2 was added the average of the Ct value for Acanthamoeba.

Editorial Office: Table S2: specify which ITS region has been amplified in qPCR.

Response to Editor Office: As Editorial Office suggested, Table S2 was changed as “the 5.8S rRNA and ITS (ITS1 and ITS2) regions”

Reviewer #1: 

Editorial Office: Line 44-45: "and that was the first-time detection in water sources, Turkey", reformulate this sentence

Response to Editor Office: As Editorial Office suggested, the sentence "and that was the first-time detection in water sources, Turkey" was edited as “and detected for the first time from water sources in Turkey.”

Editorial Office: Line 49: To "the" best of our knowledge...

Response to Editor Office: As Editorial Office suggested, the word "the" was edited as “the best of our knowledge”

Editorial Office: Line 57: "are unicellular protozoa that find commonly in soil and water" commonly find

Response to Editor Office: As Editorial Office suggested, the sentence "are unicellular protozoa that find commonly in soil and water" was edited as "are unicellular protozoa that commonly find in soil and water".

Editorial Office: Line 67: "in particularly children and young adults" immunocompetent children and young adults

Response to Editor Office: As Editorial Office suggested, the sentence "in particularly children and young adults" was edited as "in immunocompetent children and young adults”.

Editorial Office: Lines 110-112: reformulate this sentence

Response to Editor Office: As Editorial Office suggested, this sentence was reformulated as following:

“For culture and DNA isolation one liter of water sample was concentrated by filtration using a nitrocellulose membrane with a pore size of 0.22 μm.”

Editorial Office: Line 115: "It was filtrate in 0.22 μm pore-..."; write "through" instead of "in"

Response to Editor Office: As Editorial Office suggested, since figure 1 was removed upon the suggestion of other referees, the figure explain mentioned is not found in the manuscript.

Editorial Office: Line 130: Are the bacteria (E. coli) inactivated?

Response to Editor Office: As Editorial Office suggested, Yes, bacteria inactivated and added to sentence “with heat killed”.

Editorial Office: Line 136: "from" other organisms

Response to Editor Office: As Editorial Office suggested, the word “from” was added as “from other organism”

Editorial Office: Line 137: The grown "of" Acanthamoeba...was diferenciated from / Acanthamoeba trophs and cysts were diferenciated from Osmo-tolerance assay: have you axenified the Acanthamoeba samples? were they in liquid culture? Why did you add E. coli in the tolerance assay plates? The tolerance asssay could not be trustable if the authors use a bacteria suspension

Response to Editor Office: Acanthamoeba can be distinguished in NNA plates by looking at the morphological features of trophozoites and cysts. It is reported in reference sources (“Pussard M, Pons R. Morphology of cystic wall and taxonomy of genus Acanthamoeba (Protozoa, Amoebida). Protistologica. 1977;13(4):557-98 and Page FC. A new key to freshwater and soil gymnamoebae: with instructions for culture 542 1988.”). 

Acanthamoeba samples were grown on xenic culture non-nutrient agar plates by coating with heat-inactivated E. coli. Liquid PYG medium, which is axenic culture, was not used. There are no studies using liquid media in the tolerance methods of Acanthamoeba. As in many studies, the necessary nutrient E. coli is used for the proliferation of amoebae on non-nutrient agar plates. For example, reference resources: (Todd CD, Reyes-Batlle M, Martin-Navarro CM, Dorta-Gorrin A, Lopez-Arencibia A, Martinez-Carretero E, et al. Isolation and Genotyping of Acanthamoeba Strains from Soil Sources from Jamaica, West Indies. Journal of Eukaryotic Microbiology. 2015;62(3):416-21 and Caumo K, Frasson AP, Pens CJ, Panatieri LF, Frazzon AP, Rott MB. Potentially pathogenic Acanthamoeba in swimming pools: a survey in the southern Brazilian city of Porto Alegre. Annals of tropical medicine and parasitology. 2009;103(6):477-85.)

Editorial Office: Line 181: Do you know the Acanthamoeba reference strain specie?

Response to Editor Office: No, Acanthamoeba species was not identified in the study, only T4 is reported as genotype.

Editorial Office: Line 184: Center(without s) for Disease Control and Prevention

Response to Editor Office: As Editorial Office suggested, the sentence was edited as “Center for Disease Control and Prevention”.

Editorial Office: Line 187: "quantification"

Response to Editor Office: As Editorial Office suggested, the word "quantitation" was edited as “quantification”.

Editorial Office: Line 268: From the total of samples / From the 148 samples

Response to Editor Office: As Editorial Office suggested, the sentence was edited as “From the total of samples ”.

Editorial Office: Lines 282/283: ...were grown...

Response to Editor Office: As Editorial Office suggested, the word was changed as “were grown”.

Editorial Office: Table 1: positive / negative SAMPLES

Response to Editor Office: As Editorial Office suggested, table 2 was added the word “samples”

Editorial Office: As you could detect N. fowleri by qPCR, why you could not isolated it by NNA culture?

Response to Editor Office: Since Naegleria species do not differ morphologically from N. fowleri, N. fowleri was detected by qPCR from other FLA that we found positive in culture. In fact, although free-living amoebae such as Naegleria and Vermamoeba etc. have been isolated in culture, it is very difficult to diagnose directly microscopically. Therefore, the diagnosis is made with species-specific primers by PCR assay and sequence.

Editorial Office: Line 396: Reformulate this sentence

Response to Editor Office: As Editorial Office suggested, this sentence was reformulated as following: 

Line 513-515 added the sentence: “This study was occurred the quantify presence of Acanthamoeba and N. fowleri from various water sources in province İzmir, Turkey.”

Editorial Office: Lines398-406: the authors are presenting the results again. It is necessary to develope this paragraph by a comparison with other authors and reason the obtained results.

Response to Editor Office: As Editorial Office suggested, this paragraph was developed in comparison with other authors and the results are justified as follows.

Line 518-525 added the sentence: “The results of this study which indicate that the Acanthamoeba spp. occurrence in various water sources (18.24%) are similar to those results obtained in the America, Brazil, and Japan Jamaica, Iran, Thailand, Turkey. In previous reports, Naegleria spp. was found in various water sources worldwide at 0.6%–60.9% all over the world, and 0.7–10% in Turkey. [26, 35, 36]. We conclude that Naegleria spp. and Acanthamoeba spp. are free-living amoebas that have suitable growth in various water sources worldwide, but detection rates at different regions may be influenced by water types and geographical conditions.”

Editorial Office: Line 453: including rivers...

Response to Editor Office: As Editorial Office suggested, the sentence was edited as “including rivers”

Editorial Office: Lines 455-457: Reformulate this sentence

Response to Editor Office: As Editorial Office suggested, this sentence was reformulated as following:

Line 593-594 added the sentence “In this study, N. fowleri, a human pathogen, was found positive for the first time in environmental water resources collected in İzmir province, Turkey.”

Editorial Office: It could be interesting if the authors talk about the pathogenicity of the different N. fowleri genotypes.

Response to Editor Office: As Editorial Office suggested, a paragraph has been added about the effect on the pathogenicity of N. fowleri types.

Line 614-617 added the paragraph “Since there are a limited number of studies typing N. fowleri, the information about the pathogenicity of the types is insufficient. However, there is not yet conclusive evidence of any difference in virulence for any of the detected N. fowleri types. It is likely to be detected in humans, as types 2, 3, and 5 are the most common in waters.”

Editorial Office: Lines 474-475: ...,THE present study reports both the presence and THE concentration" or "presence and

concentration"..."and DEMONSTRATES THEIR RAPIDILY DETERMINATION by qPCR."

Response to Editor Office: As Editorial Office suggested, the sentence was added.

Editorial Office: Line 479: "...which are commonly detected AS CAUSAL AGENTS OF AK..."

Response to Editor Office: As Editorial Office suggested, the sentence was added as “which are commonly detected as causal agents of AK”. 

Editorial Office: Line 488: "...should increase..."

Response to Editor Office: As Editorial Office suggested, the sentence was edited as “should increase"

---

## [Editor Report · Decision Letter 1]

28 Jul 2021

PONE-D-21-09475R1

Evaluation of molecular characterization and phylogeny for quantification of Acanthamoeba and Naegleria fowleri in various water sources, Turkey

PLOS ONE

Dear Dr. Aykur,

Thank you for submitting your manuscript to PLOS ONE. After careful consideration, we feel that it has merit but does not fully meet PLOS ONE’s publication criteria as it currently stands. Therefore, we invite you to submit a revised version of the manuscript that addresses the points raised during the review process.

We look forward to receiving your revised manuscript.

Kind regards,

Maria Stefania Latrofa

Academic Editor

PLOS ONE

Journal Requirements:

Additional Editor Comments:

In my opinion the article has been improved and deserves to be published pending few suggestions

Line 36: change “which” in “with”

Lines 36-37: check the sentence; indeed the qPCR usually works on small fragments, the region that includes both 5.8S and ITS1-ITS2 could be a large fragment; so specify exactly which target gene/regions the primers are targeting.

Lines 40-41: specify which target gene has been used for the identification of Acanthamoeba isolate, see for example lines 64 and 172.

Line 78: delete “in water samples.”

Lines 277-278 and 281-282: I suggest deleting what concerns on genotypes, and possibly including this information in the specific paragraph on this topic.

Line 466: delete "a human pathogen"

---

## [Author Response · Author response to Decision Letter 1]

9 Aug 2021

Authors’ Responses to the Review Comments

Editorial Office: Please review your reference list to ensure that it is complete and correct. If you have cited papers that have been retracted, please include the rationale for doing so in the manuscript text, or remove these references and replace them with relevant current references. Any changes to the reference list should be mentioned in the rebuttal letter that accompanies your revised manuscript. If you need to cite a retracted article, indicate the article’s retracted status in the References list and also include a citation and full reference for the retraction notice.

Response to Editor Office: As Editorial Office suggested, all references have been reviewed. Some of the references from the retracted articles were removed as follows and replaced with relevant current references.

" Kolören Z, Taş B. Molecular Characterization of Acanthamoeba species in Water Resources of Ordu Province in Turkey 2017" which is a reference in the reference list [27] has been removed.

Erdogan D.D., et al., 2020, unpublished reference has been replaced with relevant current references [52 and 53]. “[52] Ertabaklar H, Turk M, Dayanir V, Ertug S, Walochnik J. Acanthamoeba keratitis due to Acanthamoeba genotype T4 in a non-contact-lens wearer in Turkey. Parasitology research. 2007;100(2):241-6. doi: 10.1007/s00436-006-0274-0. PubMed PMID: 17013653.”

“[53]. Ozkoc S, Tuncay S, Delibas SB, Akisu C, Ozbek Z, Durak I, et al. Identification of Acanthamoeba genotype T4 and Paravahlkampfia sp. from two clinical samples. Journal of medical microbiology. 2008;57(Pt 3):392-6. doi: 10.1099/jmm.0.47650-0. PubMed PMID: 18287307.”

The following sentence and cite have been removed from the manuscript. 

"Acanthamoeba T4 genotype was detected in the CSF sample of a patient with unknown cause of encephalitis in İzmir, Turkey (Aykur M., et. al., 2021, unpublished)"

“Also in previous study, we have reported the presence of N. fowleri genotype 2 from one patient in a cerebrospinal fluid sample (Aykur M., et. al., unpublished).”

Additional Editor Comments:

Editorial Office: In my opinion the article has been improved and deserves to be published pending few suggestions

Response to Editor Office: Many thanks for your time and interest in handling our manuscript. Many thanks for your insightful comments concerning our manuscript. They really helped our manuscript appear stronger. Specifically, we have revised our manuscript according to all your comments.

Editorial Office: Line 36: change “which” in “with”

Response to Editor Office: As Editorial Office suggested, line 36 “which” was changed as “with”.

Editorial Office: Lines 36-37: check the sentence; indeed the qPCR usually works on small fragments, the region that includes both 5.8S and ITS1-ITS2 could be a large fragment; so specify exactly which target gene/regions the primers are targeting.

Response to Editor Office: As Editorial Office suggested, qPCR specific primers target the ITS1 region and It has been corrected as follows. “N. fowleri positives were detected in six different water sources using qPCR with ITS regions (ITS1) specific primers”.

Editorial Office: Lines 40-41: specify which target gene has been used for the identification of Acanthamoeba isolate, see for example lines 64 and 172.

Response to Editor Office: As Editorial Office suggested, see example lines 64-172, 18S rRNA target gene has been used for the identification of Acanthamoeba isolate.

Editorial Office: Line 78: delete “in water samples.”

Response to Editor Office: As Editorial Office suggested, in line 78 “in water samples” was deleted.

Editorial Office: Lines 277-278 and 281-282: I suggest deleting what concerns on genotypes, and possibly including this information in the specific paragraph on this topic.

Response to Editor Office: As Editorial Office suggested, in lines 277-278 and 281-282 was deleted what concerns on genotypes.

Editorial Office: Line 466: delete "a human pathogen"

Response to Editor Office: As Editorial Office suggested, in line 466 “a human pathogen” was deleted.

Finally, we are sincerely grateful to both reviewers and editors for your time and interest in reviewing and accepting to publish our manuscript. We look forward to having our manuscript accepted and published with Plos one.

---

## [Editor Report · Decision Letter 2]

12 Aug 2021

Evaluation of molecular characterization and phylogeny for quantification of Acanthamoeba and Naegleria fowleri in various water sources, Turkey

PONE-D-21-09475R2

Dear Dr. Aykur,

We’re pleased to inform you that your manuscript has been judged scientifically suitable for publication and will be formally accepted for publication once it meets all outstanding technical requirements.

Kind regards,

Maria Stefania Latrofa

Academic Editor

PLOS ONE

---

## [Editor Report · Acceptance letter]

13 Aug 2021

PONE-D-21-09475R2 

Evaluation of molecular characterization and phylogeny for quantification of *Acanthamoeba* and *Naegleria fowleri* in various water sources, Turkey 

Dear Dr. Aykur:

I'm pleased to inform you that your manuscript has been deemed suitable for publication in PLOS ONE. Congratulations! Your manuscript is now with our production department. 

Kind regards, 

on behalf of

Dr. Maria Stefania Latrofa 

Academic Editor

PLOS ONE